# Psychophysiological Effects of a Single Dose vs. Partial Dose of Caffeine Gum Supplementation on the Cognitive Performance of Healthy University Students: A Placebo Controlled Study

**DOI:** 10.3390/brainsci15050536

**Published:** 2025-05-21

**Authors:** Nicolas Saavedra Velasquez, Giovanni Francino Barrera, Victor Cuadrado Peñafiel, Ricardo de la Vega Marcos

**Affiliations:** 1Department of Physical Education, Sport & Human Movement, Education Faculty, Autonomous University of Madrid, 28049 Madrid, Spain; nsaavedrav1@gmail.com; 2School of Kinesiology, Faculty of Health, Santo Tomas University, Arica 1000000, Chile; giovannifrancinoba@santotomas.cl

**Keywords:** alertness, caffeine, cognitive performance, exercise, psychophysiological

## Abstract

**Background:** Caffeine has become the psychostimulant with the highest use worldwide by different segments of the population. This is mainly due to the wide variety of benefits it offers in different contexts of use. It is available in various forms, with caffeine chewing gum recently generating great interest due to its characteristics and absorption time. **Methods:** A placebo-controlled study was conducted in which 20 healthy university students were exposed to three different conditions (single dose, partial dose, and placebo). The intervention consisted of a dual task in which heart rate, perceived exertion, and reaction time were monitored using the Stroop test and choice reaction time test while participants performed two blocks of cycloergometer exercise. **Results:** A *t*-test comparison between blocks showed differences in the Stroop test under all conditions, with the single dose having the best performance (Gr.A *p* < 0.001; Gr.B *p* < 0.029; Gr.C *p* < 0.009). The single dose group also showed favorable results for the HR/RPE ratio (*p* < 0.044) and an increase in the rate of perceived exertion (*p* < 0.006). No changes in reaction time were observed under any condition of the choice reaction time test. **Conclusions:** These results suggest that caffeine supplementation has positive effects on variables related to psychophysiological performance during a dual task. A single dose showed the best results in this study; however, longer intervention designs could be considered in the future to see the effect of partial doses of caffeine over time.

## 1. Introduction

Caffeine has established itself as one of the most widely used stimulants in the population. Currently, it is probably the most widely used pharmacological substance in the world due to its easy access and the various contexts in which it can be used [1]. As discussed in a recent review [2], its use is mainly due to its effects on the central nervous system (CNS) [3], where its main mechanism of action is the blockade of adenosine receptors (A1 and A2A) [4]. It is possible to find more isoforms of these receptors (A1, A2A, A2B, and A3), which are responsible for mediating different effects, where A1 has a mainly inhibitory effect, while A2A is related to excitatory effects [5].

Blocking these receptors has been shown to have beneficial effects on neuromuscular variables such as muscle power and strength [6,7]. Caffeine has been shown to increase calcium release from the sarcoplasmic reticulum [8] and also to play a role in regulating plasma and intracellular potassium (K) levels by stimulating the Na-K pump [9]. Therefore, these phenomena may contribute to improved neuromuscular performance, but the mechanisms are not yet fully understood.

In this sense, it should be noted that the benefits of caffeine consumption are not only related to physical performance, but also play an important role in improvements related to cognitive functions [10,11]. Various studies have consistently shown that caffeine improves variables related to cognitive function, mood, level of wakefulness, alertness, delayed onset of fatigue, and reaction time, among others [12,13,14,15,16]. For these reasons, the use of caffeine has generated a great deal of interest among college students seeking to optimize their academic performance [17,18,19].

Traditionally, caffeine has been administered in both research and sports contexts through capsules, caffeinated liquids, and energy drinks, but it is also available in other forms such as gels, bars, chewing gum, mouth sprays, nasal sprays, and mouthwashes, which can affect how quickly it is absorbed into the blood from the oral and intestinal mucosa [20]. In this sense, and due to the faster absorption through the oral cavity, chewing gum is considered to be a faster way to absorb the stimulant than drinks and capsules [21]. In general, studies using this form have shown that the administration times range from 5 to 10 min of chewing [22,23,24,25,26,27]. In this sense, it has been shown that about 80% of caffeine is released with a 5 min chewing protocol, so it is suggested that a 10 min protocol is probably appropriate to maximize release and minimize variation in the doses administered [28].

Psychophysiology is defined as the study of the relationship between physiological signals recorded by the body and the brain and cognitive and emotional processes [29]. This relationship is modified in response to different situations and/or contexts of stress, and the variability of this psychophysiological response is also conditioned by the magnitude of the stressor (context, intensity of exercise, complexity of a task, among others). When the magnitude of stress is low, an individual can focus more on sensory signals and/or external stimuli. In contrast, when the stress magnitude is high, attention is focused mostly on internal signals, thus inducing a greater impact on the psychophysiological response.

People are constantly exposed to stressful situations, either acute, such as daily workload, sports competition, and/or pressure situations, or chronic, in the case of neurodegenerative diseases. In any case, these situations induce a variation in the psychophysiological response that leads to deterioration and fatigue, both physical and cognitive. Cognitive fatigue is a phenomenon that can be conditioned by a variety of factors, which can result in, among other things, a decrease in alertness, concentration, motivation, and work productivity [30].

Despite the accumulating evidence for the beneficial effects of caffeine on physical and cognitive performance, there is a paucity of studies that comparatively analyze the psychophysiological effects of different dosing schedules (single versus partial) of caffeine in gum form during dual tasks involving simultaneous physical exertion and cognitive load. Furthermore, most studies focus on traditional delivery forms (capsules, beverages) and not on rapidly absorbed forms such as chewing gum, which is a major limitation. This research aims to fill this gap by providing comparative evidence under controlled conditions using standardized cognitive tasks in a moderate physical exertion environment.

It has been shown that physical exercise has an impact on the psychophysiological response, which could be increased and/or attenuated together with caffeine supplementation. Therefore, the objective of this study is to determine the effects of caffeine gum supplementation with different administration times on variables related to cognitive performance in healthy university students during a dual task.

## 2. Materials and Methods

### 2.1. Participants

A statistical power analysis (GPower 3.1) was performed to determine the sample size, with an effect size of 0.6, alpha value of 0.05, and power of 0.8, resulting in a sample size of N = 24. In total, 24 healthy male university students were recruited, 20 of whom participated in the intervention design (characteristics shown in Table 1). Recruitment was done through dissemination among the students of the Universidad Santo Tomas, Arica, Chile. To allow their participation, the following criteria were defined: between 18 and 35 years of age, no underlying pathologies that prevent physical exercise, and not habitual consumers of caffeine. Each participant underwent an electrocardiogram, which was reviewed by a member of the university medical team to approve participation in the study. A within-subject design was used in which each participant underwent three experimental conditions.

### 2.2. Experimental Design

A single-blind, cross-over, within-subject, repeated-measures design was used for this study. Each participant performed in all experimental groups (Gr.), the order of which was randomly assigned by drawing lots. In the first instance, an incremental test was performed to determine the maximum oxygen consumption and to establish the workload during the experimental sessions, a questionnaire on caffeine consumption was completed, and familiarization with cognitive tests was carried out. They were asked not to consume caffeine in any form for 24 h before the intervention and to avoid any strenuous physical activity for the last 48 h. Each participant then completed 3 experimental conditions separated by at least 7 days. The single-dose condition (Gr.A) corresponded to 400 mg of caffeine in block 1 and 0 mg in block 2; the partial-dose condition (Gr.B) corresponded to 200 mg of caffeine in both blocks; and the placebo condition (Gr.C) corresponded to 0 mg of caffeine in both blocks. At each laboratory visit, the order of tests was as follows: choice reaction time test; administration of caffeine gum according to the corresponding condition; after 10 min of chewing gum, the first period of work began on a cycle ergometer at 60% Wmax together with the performance of the Stroop test; heart rate (HR) and rate of perceived exertion (RPE) measurement; a 10 min break, during which gum was again administered; and a second period of the cycle ergometer at 60% Wmax together with the performance of the Stroop test, HR and RPE measurement, and choice reaction time test (Figure 1). The descriptive analysis of the variables is shown in Table 2.

### 2.3. Caffeine Supplementation

The administration was in the form of Military Energy Gum (MEG) brand chewing gum, with each piece containing 100 mg of caffeine. The participants were unaware of the amount of caffeine administered, as caffeine-free gum of the same size, color, and flavor was used to match the total amount of gum administered after each condition. In addition, the participants were blindfolded at the time of administration, and the gum was delivered in a glass so that they had no visual and tactile information about the dose administered. According to the characteristics of the participants and considering their average weight, the amount of caffeine administered was within the ranges recommended in the literature to increase its ergogenic effect (3–6 mg/kg) [2,11].

### 2.4. Maximum Oxygen Consumption

Maximum oxygen consumption was determined using a Monark model 827 cycloergometer (Monark, Varberg, Sweden) and a MetaLyzer 3B model CORTEX ergospirometer (CORTEX, Leipzig, Germany). The protocol consisted of a two-minute warm-up at 25 W at a cadence of 50 rpm, followed by two-minute blocks at 50 rpm with progressive increases of 25 W to exhaustion. To determine VO2max, at least two of the following variables were used as criteria: achieving a respiratory exchange index (RIR) ≥ 1.1, a flattening of VO2, and a heart rate ≤ 10 beats/min or ≤ 5% of the age-estimated maximum (220—age). All cardiorespiratory variables were controlled in real time for each breath using MetaSoft Studio software (Version 5.9.0-4.1.0), which was compatible with the ergospirometer.

### 2.5. Cognitive Task

In this study, cognitive performance was assessed using two validated tests: the choice reaction time test, which measures alertness and motor response speed, and the Stroop Test, which assesses cognitive flexibility, selective attention, and psychomotor speed in the face of conflicting stimuli.

#### 2.5.1. Choice Reaction Time Test

The test provides a cognitive performance index that measures alertness and motor performance. This assessment of complex reaction time was performed using the Milisecond program (Inquist 7). The test consists of pressing a Home button until one of the five squares in different positions lights up. The subject must quickly touch the randomly assigned square and return to the Home button as quickly as possible (Figure 2). The mean of the response times was used for the subsequent analysis.

#### 2.5.2. Stroop Test

The Stroop test is an assessment of cognitive flexibility and psychomotor speed. The participants completed it on a tablet using the EncephalApp Stroop app (Version 2.1.0). Each participant was shown a series of discordant stimuli in which they had to touch the color of the word presented on the screen. Each word that said a particular color was presented in a discordant color (Figure 3). The mean of the response times and the number of errors were used for the subsequent analysis.

### 2.6. Data Analysis

Statistical analysis was performed using SPSS Statistics 29 for MacOS. To determine the distribution of the sample, the variables were analyzed using the Shapiro–Wilk test. Based on the Shapiro–Wilk results, the results of the different variables are presented in two different tables (one for each type of distribution). A *t*-test for dependent samples was used to determine the differences between blocks in the three different conditions, while a repeated-measures ANOVA with Bonferroni adjustment for multiple comparisons was used for the analysis between the three conditions. Wilcoxon and Friedman tests were used for the aforementioned nonparametric variables. In all calculations, *p* < 0.05 was considered statistically significant.

## 3. Results

The descriptive analysis is presented in Table 2, and the analysis of the different variables evaluated is presented in Table 3 and Table 4, according to the type of distribution that each of them had (parametric and non-parametric).

### 3.1. Cognitive Task

#### 3.1.1. Choice Reaction Time Test

No differences were found either in the comparison between the blocks or in the ANOVA for the comparison between the conditions.

#### 3.1.2. Stroop Test

Regarding performance on the Stroop test, the analysis of t-tests showed differences in the mean Stroop response in all conditions (Gr.A *p* < 0.001; Gr.B *p* < 0.029; Gr.C *p* < 0.009). No changes were found in the number of errors in any of the conditions (Gr.A *p* > 0.887; Gr.B *p* > 0.733; Gr.C *p* > 0.378). No differences were found when comparing the conditions using ANOVA.

### 3.2. Rate of Perceived Exertion and Heart Rate

After the respective analysis of the *t*-tests between blocks 1 and 2, changes were found in the variable RPE in the Gr.A condition (*p* < 0.006) and in HR/RPE also in the Gr.A condition (*p* < 0.044). No differences were found when comparing the conditions using ANOVA.

## 4. Discussion

Considering that physical exercise can influence psychophysiological responses and that this effect could be modulated by caffeine supplementation, this study was designed to investigate how different caffeine chewing gum administration strategies (single and partial doses) affect cognitive performance under dual-task conditions. The results will allow us to assess the extent to which these dosing strategies fulfill the proposed potential and whether they offer practical benefits in the context of cognitive and physical stress.

In general, in variables related to cognitive performance, caffeine consumption has been shown to decrease reaction time and increase alertness and attention [31,32]. However, the results obtained in this study using the choice reaction time test did not show significant differences between blocks in any of the conditions. It is important to note that the means of the second block in both caffeine conditions were lower than the means of the first block (Gr.A: 558.09 vs. 553.79; Gr.B: 570.56 vs. 568.15), but the placebo condition was the only one to show an increase in the second block (Gr.C: 560.43 vs. 569.02). These results suggest that although there is no statistically significant difference, caffeine supplementation seems to attenuate the loss of response speed in both conditions (Gr.A and Gr.B), and even in both, being lower in the second blocks.

The Stroop test showed significant changes in the analysis between blocks in the three conditions evaluated, with the single dose condition showing the greatest difference (*p* < 0.001). It should be noted that the placebo condition also showed changes despite the absence of caffeine. Similar results have been found in other studies, where the expectation of improvement can increase performance despite a placebo [33,34]. It is likely that there was a learning effect in the subjects who showed positive results on the Stroop response times in the placebo condition. Although no significant differences were found in the number of errors made during the Stroop test, it was possible to see that there was variability in the response between conditions, with the single dose being the one that resulted in the fewest errors (Gr.A *p* > 0.887; Gr.B *p* > 0.733; Gr.C *p* > 0.378).

On the other hand, with respect to variables related to physiological response, the HR results showed no significant changes in any condition. This result is similar to studies where the use of caffeinated energy drinks also showed no differences in this variable [35,36], but it is important to note that despite there being no statistical change, HR was higher in the placebo condition in both blocks.

In a situation of physical/cognitive stress, it is normal for the RPE to tend to increase due to the fatigue associated with the activity. It has been shown that caffeine supplementation can attenuate and even reduce the RPE during [37] and after exercise [38], but these results are not related to those found in this study. No condition showed a reduction in the RPE, but contrary to what was expected, caffeine supplementation showed an increase in the single dose condition (*p* < 0.006).

In this sense, although all conditions showed an increase in the RPE in block 2, only the single-dose condition showed significant changes (*p* < 0.006), but it was also the condition with the lowest mean of the three in block 1. The placebo condition had the highest mean in both blocks. These results provide indications on the behavior of this variable in the three different conditions, where, although there were no significant changes, it could be deduced that in activities and/or situations of reduced time, supplementation with a single high dose could be an effective alternative to improve the RPE. The alternative of partial doses of caffeine could be interesting to apply in contexts and/or situations of more prolonged stress; however, regarding the results obtained, it is necessary to test this in studies that can integrate an intervention design with longer durations. The HR/RPE ratio is a variable that can be used as a practical measure to monitor fatigue levels [39]. This ratio provides relevant information about the subject’s psychophysiological response and ability to manage effort. The changes presented in the single-dose condition (*p* < 0.044) indicate that the subjects had a better ability to withstand the effort induced by the experimental design.

### 4.1. Limitations of the Study and Future Guidelines

It was difficult to administer caffeine based on body weight, since the gum contained a fixed dosage of 100 mg per unit, so the participants did not receive the same relative amount depending on their body weight. It is recommended that future studies using this form of administration should consider adjusting the amount of caffeine in each gum in relation to body mass to ensure a more accurate dose and reduce inter-individual variability in response. As mentioned above, another variable that is important to consider is the possible learning effect in subjects who did not have a placebo as their first assigned condition, which may allow them to perform better in subsequent conditions. This is a problem that occurs in this type of intervention and in different types of studies, which is why the assignment of conditions was randomized to minimize this situation.

It would be interesting to study intervention designs using longer durations to determine whether there are greater benefits from using partial doses over time. Caffeine supplementation has also been consistently shown to have positive effects on physical performance [40,41,42,43,44], so it would be interesting to replicate this design in the future by including variables related to time to exhaustion, strength, and power, among others.

Regarding possible adverse effects due to caffeine consumption, no participant reported gastrointestinal discomfort or insomnia problems. Only in a few cases was an increased state of alertness reported, but this did not cause any discomfort; however, these responses were not reported in any of the questionnaires. Therefore, it would be important for future research to be able to report possible adverse effects in this way.

A limitation of the present study is the lack of a no-intervention control group, which makes it difficult to completely rule out the influence of learning effects or expectancy biases, particularly in the placebo condition. Although the observed cognitive improvements in the placebo group may reflect true placebo responses, it is also possible that they were partly due to practice effects associated with repeated testing. Future studies should consider including a no-intervention control group or using a crossover design with appropriate washout periods to better isolate the specific effects of caffeine supplementation from non-specific cognitive gains.

### 4.2. Practical Applications

It is important to consider and exercise caution when reviewing the results because although certain variables did not reach statistical significance, they do provide us with relevant information about trends in response to caffeine supplementation, which may be useful for decision making in certain types of contexts. A single high dose of caffeine (400 mg) may be recommended for improvement on tests of cognitive ability, but as noted above, studies with longer-term designs are needed to determine whether partial doses could provide a longer effect.

## 5. Conclusions

In conclusion, the results obtained in this study indicate that caffeine gum supplementation can improve certain indicators of psychophysiological performance during dual tasks combining physical exertion and cognitive demand, especially when a single dose of 400 mg is used. This strategy showed significant positive effects on Stroop test response time and the HR/RPE ratio, suggesting improved tolerance to perceived exertion and greater psychophysiological efficiency.

Although no significant improvements were observed in the choice reaction time test, the results suggest that caffeine may help alleviate cognitive fatigue and improve the ability to maintain performance under physical stress, which is highly relevant both in sports contexts where athletes must make quick and accurate decisions under conditions of fatigue and in occupational contexts (health personnel), military, and sectors where there is high cognitive and physiological demand.

The results also highlight the importance of dosing strategy. The single dose showed greater efficacy in short-duration efforts, while partial doses could be the subject of future studies to evaluate their usefulness in contexts of prolonged or intermittent efforts. This information may be of practical use in sports disciplines or other contexts that require sustained concentration and/or moderate physical effort.

Finally, we highlight the need for future studies with longer intervention designs, the inclusion of physical variables such as strength or power, and more detailed monitoring of potential side effects to strengthen and expand the safe and effective application of caffeine in different use scenarios.

## Figures and Tables

**Figure 1 brainsci-15-00536-f001:**
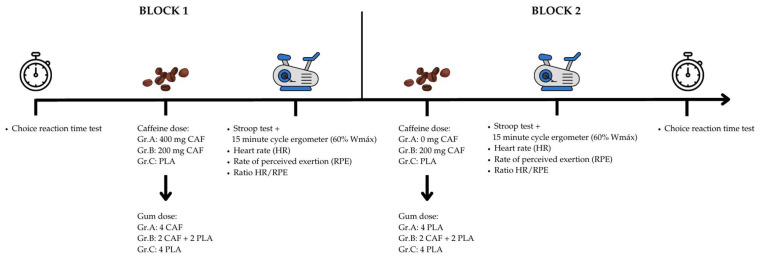
Experimental design. Group (Gr.); caffeine (CAF); placebo (PLA).

**Figure 2 brainsci-15-00536-f002:**
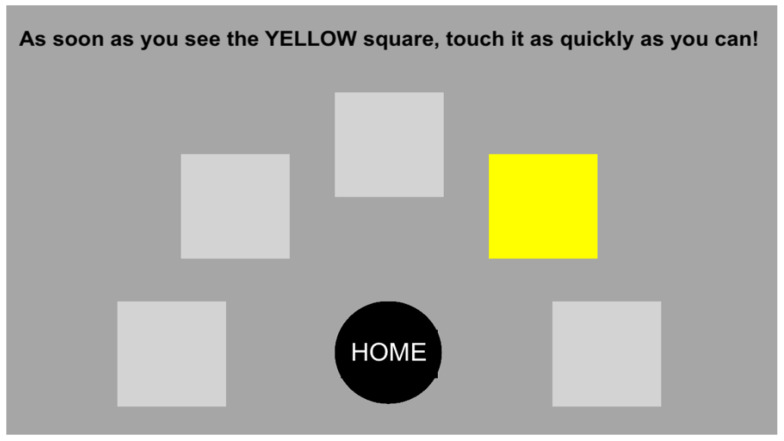
Choice reaction time test: visual representation.

**Figure 3 brainsci-15-00536-f003:**
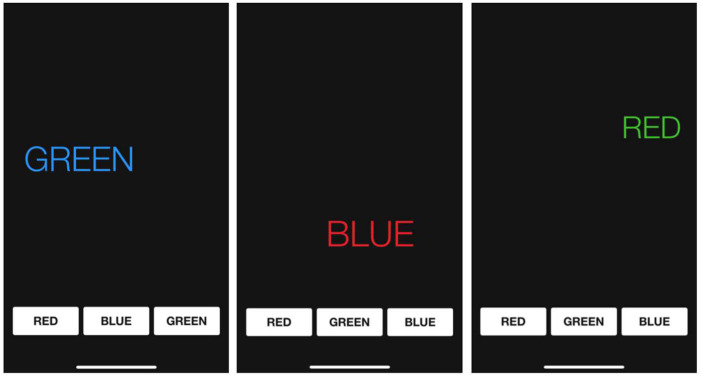
Stroop test: visual representation.

**Table 1 brainsci-15-00536-t001:** Physical characteristics of the subjects.

Physical Characteristics	Participants (n = 20)
Age (years)	21.8 ± 3.1
Height (cm)	174.1 ± 7.2
Weight (kg)	78.3 ± 14.6
BMI	25.8 ± 4.3
Vo2max (mL/kg/min)	34.2 ± 6.3

**Table 2 brainsci-15-00536-t002:** Descriptive analysis.

		N	Mean	Std. Deviation
Heart rate, Gr.A	Block 1	20	130.35	18.18
	Block 2	20	135.65	18.18
Heart rate, Gr.B	Block 1	20	133.45	15.99
	Block 2	20	135.50	12.66
Heart rate, Gr.C	Block 1	20	137.35	14.93
	Block 2	20	137.70	13.45
Rate of perceived exertion, Gr.A	Block 1	20	4.50	1.76
	Block 2	20	5.15	1.93
Rate of perceived exertion, Gr.B	Block 1	20	4.95	1.90
	Block 2	20	5.10	1.86
Rate of perceived exertion, Gr.C	Block 1	20	5.15	1.73
	Block 2	20	5.35	1.98
Ratio HR/RPE, Gr.A	Block 1	20	35.99	24.09
	Block 2	20	32.06	17.74
Ratio HR/RPE, Gr.B	Block 1	20	32.65	16.96
	Block 2	20	30.98	13.38
Ratio HR/RPE, Gr.C	Block 1	20	30.47	12.61
	Block 2	20	31.40	17.23
Choice reaction time test (ms), Gr.A	Block 1	20	558.09	84.57
	Block 2	20	553.79	73.54
Choice reaction time test (ms), Gr.B	Block 1	20	570.56	99.24
	Block 2	20	568.15	96.82
Choice reaction time test (ms), Gr.C	Block 1	20	560.42	90.65
	Block 2	20	569.02	93.73
Stroop mean response (s), Gr.A	Block 1	20	1.15	0.17
	Block 2	20	1.06	0.13
Stroop mean response (s), Gr.B	Block 1	20	1.11	0.17
	Block 2	20	1.07	0.18
Stroop mean response (s), Gr.C	Block 1	20	1.10	0.14
	Block 2	20	1.04	0.13
Errors, Gr.A	Block 1	20	1.20	1.40
	Block 2	20	1.25	1.55
Errors, Gr.B	Block 1	20	1.50	1.28
	Block 2	20	1.70	1.42
Errors, Gr.C	Block 1	20	1.35	1.42
	Block 2	20	1.25	1.37

**Table 3 brainsci-15-00536-t003:** Analysis between variable blocks with parametric distribution.

	Mean (B1-B2)	Std. Deviation	Std. Error Mean	t	gl	g	*p*
Heart rate, Gr.A	−5.30	18.01	4.03	−1.32	19	18.77	0.20
Heart rate, Gr.B	−2.05	10.02	2.24	−0.91	19	10.44	0.37
Heart rate, Gr.C	−0.35	6.75	1.51	−0.23	19	7.03	0.82
Rate of perceived exertion, Gr.A	−0.65	0.93	0.21	−3.11	19	0.97	<0.01 **
Rate of perceived exertion, Gr.B	−0.15	0.88	0.20	−0.77	19	0.91	0.45
Rate of perceived exertion, Gr.C	−0.20	1.11	0.25	−0.81	19	1.15	0.43
Choice reaction time test, Gr.A	4.30	54.21	12.12	0.35	19	56.47	0.73
Choice reaction time test, Gr.B	2.41	43.15	9.65	0.25	19	44.96	0.81
Choice reaction time test, Gr.C	−8.60	37.41	8.37	−1.03	19	38.97	0.32
Stroop total response, Gr.A	0.10	0.08	0.02	5.60	19	0.08	<0.001 ***
Stroop total response, Gr.B	0.04	0.08	0.02	2.37	19	0.09	0.03 *
Stroop total response, Gr.C	0.05	0.08	0.02	2.89	19	0.08	0.01 **

Mean (B1-B2): values presented between block 1 and block 2 comparison in each variable; * *p* < 0.050; ** *p* < 0.010; *** *p* < 0.001.

**Table 4 brainsci-15-00536-t004:** Analysis between variable blocks with non-parametric distribution (Wilcoxon).

	Z	r	*p*
Ratio HR/RPE, Gr.A	−2.01 ^b^	−0.32	0.04 *
Ratio HR/RPE, Gr.B	−0.99 ^b^	−0.16	0.32
Ratio HR/RPE, Gr.C	−0.21 ^b^	−0.03	0.84
Errors, Gr.A	−0.14 ^d^	−0.02	0.89
Errors, Gr.B	−0.34 ^d^	−0.05	0.73

* *p* < 0. 050. b—Based on positive ranks. d—Based on negative ranks.

## Data Availability

The data presented in the study are available upon request from the corresponding author. They are not publicly available due to ethical and data protection restrictions that currently prevent their disclosure.

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
