# Peer review of "Psychophysiological Effects of a Single Dose vs. Partial Dose of Caffeine Gum Supplementation on the Cognitive Performance of Healthy University Students: A Placebo Controlled Study"

_brainsci, 2025, doi:10.3390/brainsci15050536_

Round 1

Reviewer 1 Report

Comments and Suggestions for Authors

Review

General Comments: The present article investigates the effects of caffeine gum administered at different time intervals on cognitive performance. The topic is interesting and appears to contribute some novel aspects to the existing literature. Additionally, the manuscript appears to be well written and structured; however, it lacks certain details that I consider relevant. Firstly, there is a need to align the sections of the manuscript with the STROBE checklist recommendations for cross-sectional studies. Additionally, once the manuscript has been adjusted, the STROBE checklist should be included as supplementary material.

Title: Psychophysiological Effects: Single Dose vs Partial Dose of Caffeine Gum Supplementation on the Cognitive Performance of Healthy University Students. As recommended by the STROBE checklist, it is necessary to indicate the type of study in the title. Indicate that this is a placebo-controlled study.

Abstract: It is suggested that the research gap be more clearly articulated in the abstract. Include the type of study in the Methods section. Improve the description of the results.

Introduction: Although the introduction is well written and presents a good flow of ideas, it does not clearly articulate the research problem. What is the gap? What is truly lacking in the current literature? These aspects should be made more explicit. Additionally, a research hypothesis should be proposed and included after the objectives. Furthermore, it is not clear what the test variable will explore. Cognitive performance can be assessed in a number of ways, and this should be clearly defined.

Methods:  As previously mentioned, there is a need to revise the Methods section in accordance with the STROBE guidelines to enhance transparency and reproducibility. Cite the STROBE guidelines in the initial section of the Methods to indicate adherence to standardized reporting for cross-sectional studies.

Include information regarding the ethics committee approval number. Additionally, provide details on how participant recruitment was conducted.

“All subjects who did not meet the inclusion criteria (approval by the University medical board after an electrocardiogram) were excluded”: this information is unclear and lacks coherence. Clearly present the inclusion and exclusion criteria. Also, include the sample size calculation used to ensure adequate statistical power.

There is a strong need to include a dedicated section describing the randomization process. How exactly was it conducted?  As an example, observe how this was presented in a specific article from the field: DOI: 10.3390/nu17071261

Indicate in the figure legend what Gr, PLA and CAF stand for. I strongly suggest consistently using these abbreviations throughout the manuscript.

Shouldn't the table of descriptive analyses be presented in the Results section?

There is a strong need to include detailed information regarding the placebo condition. Create a dedicated section for this purpose. What was the composition of the placebo? Were its physical characteristics, such as texture and appearance, identical to the active intervention? Were the packaging and administration methods similar? Was blinding or double-blinding implemented? If blinding was employed, provide a thorough explanation of how it was conducted, covering all relevant procedural details.

Additionally, it is clear that the use of fixed caffeine doses introduces a significant source of bias. To minimize this issue, I suggest including in your Results section data concerning the relationship between caffeine dose and body mass. Refer to the article I previously mentioned as an example of how to approach this analysis.

If each piece of gum contained 100 mg of caffeine, participants were likely aware of the amount being administered. Explain how this was handled in the study, was the dosage disclosed to participants? If so, under what circumstances? If not, describe the strategies used to blind participants to the caffeine content, ensuring the integrity of the placebo-controlled design.

Regarding the statistical analysis, I suggest providing more precise information. Specify which variables met the assumptions of normality and were analyzed using analysis of variance, and which did not. Additionally, include 95% confidence interval data, as required by the STROBE guidelines. If an ANOVA was used, why was the effect size of the intervention not calculated?

Results: To begin the Results section, it is necessary to indicate which variables met the assumption of normality and which did not.

How was the recruitment process conducted? What were the inclusion and exclusion criteria, and how was sample loss handled? Initially, how many participants were targeted for enrollment? How many failed to complete the study visits? How many were lost due to personal reasons? Please begin the Results section by presenting a flowchart. Although it is not explicitly required by the STROBE checklist, including such a diagram greatly enhances the transparency of the research process.

There is no mention of primary or secondary outcome variables. Reassess the Methods section in accordance with the STROBE checklist, and structure the Results section by first presenting the primary outcome variable, followed by the secondary outcomes. Without clearly defining these aspects, it is not possible, for example, to properly justify the sample size calculation.

I'm sorry, but I am unable to understand your Table 3. What does a mean heart rate of –5.3 represent? If I am confused, it is likely that others will be as well. Present the data as it is, rather than as the mean of the differences. It is not possible to interpret the results accurately without access to the raw dataset.

As previously mentioned, there is a lack of alignment in your manuscript. You begin the Results section with heart rate and RPE, but what is your primary outcome? Shouldn't it be cognitive performance? Moreover, the corresponding section appears only at the end. I strongly recommend that you revise and clarify the sections on RPE and cognitive performance. Include measures of central tendency and dispersion, 95% confidence intervals, and effect sizes to ensure clarity and rigor in the presentation of your findings.

Additionally, among the participants in your sample, were all of them responders to caffeine? And what about potential placebo responders? This is a common phenomenon observed in research studies. Please include in your Results section a breakdown of responders and non-responders to the intervention, clearly presenting this information in a structured and interpretable format.

Please include, in a final subsection of the Results, any potential side effects, even if minimal. For example, did participants who ingested caffeine experience any noticeable autonomic changes? Did anyone report gastrointestinal discomfort? Present this information in a table showing the percentage of occurrence for each reported effect.

Discussion: Begin your Discussion section by restating your research objective, rather than immediately comparing findings. Refer to the STROBE checklist, which recommends clearly reiterating the study purpose at the start of the discussion to frame the interpretation of results.

Structure your Discussion section based on the definition of your outcome variables. Begin by addressing your primary outcomes, followed by the secondary ones, ensuring that your interpretations and comparisons are clearly aligned with the study's objectives and statistical findings.

There is clearly a need to expand the Discussion section further. Please reorganize and expand the Discussion section. Present potential underlying mechanisms, even if by inference, to help contextualize and support the interpretation of your findings within the broader scientific literature.

Were the variables not statistically different due to insufficient statistical power? This potential source of bias could be mitigated if such information were provided.

Your Practical Applications section is overly limited. Furthermore, the following statement should not be included in this section: “It is important to consider and exercise caution when reviewing the results, because although certain variables did not reach statistical significance, they do provide us with relevant information about trends in response to caffeine supplementation, which may be useful for decision making in certain types of contexts”.

Conclusion: I am not certain that the manuscript can be concluded in its current form. I suggest revising the Results section to ensure greater clarity; once that is accomplished, the Conclusions should be reassessed accordingly.

Author Response

Por favor vea el archivo adjunto.

Reviewer 2 Report

Comments and Suggestions for Authors

The manuscript by Velásquez et al entitled "Psychophysiological Effects: Single Dose vs Partial Dose of Caffeine Gum Supplementation on the Cognitive Performance of Healthy University Students” investigated the effects of single-dose (400 mg) vs. partial-dose (200 mg x 2) caffeine gum on cognitive performance in 20 healthy male university students during a dual-task protocol (cycling + Stroop/choice reaction time tests). Using a single-blind, crossover design, participants completed three conditions: single-dose, split-dose, and placebo. Although the manuscript provides interesting data, a few limitations must be thoroughly addressed. These critiques highlight gaps that, if addressed, would significantly strengthen the study’s validity and impact. The authors should consider these points in revisions or future research.

Major Drawbacks

  • Sample Size and Demographics: The study included only 20 male university students, which limits generalizability. The authors are advised to increase sample size and include diverse demographics (e.g., females, different age groups, varying fitness levels) to enhance external validity.
  • Lack of Control for Caffeine Habituation: Participants' habitual caffeine intake was not controlled, which could influence results due to tolerance effects. It is highly recommended to measure and adjust for baseline caffeine consumption or exclude habitual users to minimize confounding.
  • Placebo Effect and Learning Bias: The placebo condition showed improvements, likely due to learning effects or expectation bias. The authors should include a control group without any intervention or use a crossover design with washout periods to account for practice effects.
  • Dose Standardization: Caffeine was administered in fixed amounts (100 mg/gum) rather than adjusted for body weight, potentially leading to variable effects. It would be ideal for authors to standardize doses by body weight (e.g., mg/kg) to ensure consistent physiological impacts.
  • Limited Cognitive Measures: Only two cognitive tests (Stroop, Choice Reaction Time) were used, which may not capture broader cognitive effects. The authors should include additional tests (e.g., memory, and attention spans) to comprehensively assess cognitive performance.
  • Short-Term Design: The study lacked long-term follow-up, limiting insights into the sustained effects of partial dosing. It would be logical to extend the intervention period or include multiple testing sessions to evaluate temporal effects.

Minor Drawbacks

  • Heart Rate Variability (HRV) Missing: HRV, a robust marker of psychophysiological stress, was not measured. The authors should be incorporating HRV analysis to better understand autonomic nervous system responses.
  • Subjective Measures: RPE (Rate of Perceived Exertion) is subjective and prone to bias. It is highly suggested to complement RPE with objective measures like lactate threshold or electromyography (EMG).
  • Gum Administration Protocol: The chewing duration (10 minutes) was fixed, but individual chewing efficiency may vary. Monitor saliva caffeine levels to confirm absorption consistency.
  • Statistical Power: A small sample size may underpower the study to detect subtle effects. Conduct a power analysis post-hoc or a priori to justify sample size.
  • Lack of Blinding Verification: The single-blind design may not fully prevent participant bias. The authors should use/perform double-blinding and assess blinding success via post-study questionnaires.
  • Incomplete Reporting of Statistical Assumptions: The manuscript does not mention corrections for multiple comparisons (e.g., Bonferroni) or effect sizes for non-significant results. Report effect sizes (e.g., Cohen’s *d*) and adjust *p*-values for multiple tests to reduce Type I errors.

Reviewer 3 Report

Comments and Suggestions for Authors

The current manuscript reported the effects of using single dose vs partial dose of caffeine gum supplementation on healthy university students. Here are some comments to be addressed before a decision is made.

  1. Kindly add the inclusion and exclusion criteria in the abstract
  2. I can find only 20 subjects not enough to give conclusive results
  3. I suggest adding a list of abbreviations at the end of the article
  4. The conclusion section is too short so modify accordingly
  5. The presence of study limitation is a merit to this article

Round 2

Reviewer 1 Report

Comments and Suggestions for Authors I observed the article and I believe that the researchers achieved what was necessary for publication. Although they do not cover 100% of the items listed, the manuscript has been improved.   Thanks

Author Response

Por favor vea el archivo adjunto.

Reviewer 2 Report

Comments and Suggestions for Authors

1. Inconsistent Statistical Reporting: Tables 3 and 4 report Hedges’ *g* and *r* values inconsistently (e.g., some non-significant results lack these metrics). Standardize reporting for all comparisons, including effect sizes for non-significant results to enhance transparency.

2. Figure 1 Legend: Abbreviations (Gr., CAF, PLA) are undefined in the legend. Define all abbreviations in the legend or main text.

3.Discussion Flow: HR/RPE and Stroop results are fragmented across sections. Reorganize by outcome (cognitive vs. physiological) for coherence.
